# The mitochondrial permeability transition pore activates the mitochondrial unfolded protein response and promotes aging

Suzanne Angeli[1]*, Anna Foulger[1], Manish Chamoli[1], Tanuja Harshani Peiris[1], Akos Gerencser[1], Azar Asadi Shahmirzadi[1,2], Julie Andersen[1,2]*, Gordon Lithgow[1,2]*

[1]Buck Institute for Research on Aging, Novato, United States; [2]USC Leonard Davis School of Gerontology, University of Southern California, Los Angeles, United States

**Abstract** Mitochondrial activity determines aging rate and the onset of chronic diseases. The mitochondrial permeability transition pore (mPTP) is a pathological pore in the inner mitochondrial membrane thought to be composed of the F-ATP synthase (complex V). OSCP, a subunit of F-ATP synthase, helps protect against mPTP formation. How the destabilization of OSCP may contribute to aging, however, is unclear. We have found that loss OSCP in the nematode *Caenorhabditis elegans* initiates the mPTP and shortens lifespan specifically during adulthood, in part via initiation of the mitochondrial unfolded protein response (UPR[mt]). Pharmacological or genetic inhibition of the mPTP inhibits the UPR[mt] and restores normal lifespan. Loss of the putative pore-forming component of F-ATP synthase extends adult lifespan, suggesting that the mPTP normally promotes aging. Our findings reveal how an mPTP/UPR[mt] nexus may contribute to aging and age-related diseases and how inhibition of the UPR[mt] may be protective under certain conditions.

*For correspondence:
sangeli@buckinstitute.org (SA);
jandersen@buckinstitute.org (JA);
glithgow@buckinstitute.org (GL)

**Competing interests:** The authors declare that no competing interests exist.

## Introduction

As mitochondrial function declines with age, the frequency of the mitochondrial permeability transition pore (mPTP) increases (*Rottenberg and Hoek, 2017*). The mPTP is central to early-stage pathologies associated with several age-related diseases, including Alzheimer's disease (AD) and Parkinson's disease (PD) and late-stage pathologies of ischemia-reperfusion injuries including heart attack and stroke (*Ong et al., 2015*; *Panel et al., 2018*). The mPTP is a pathological channel that forms in the inner mitochondrial membrane in response to excessive cytosolic $Ca^{2+}$ or high ROS conditions. Sustained opening of the mPTP leads to outer mitochondrial membrane rupture, release of $Ca^{2+}$ into the cytosol, and cell death (*Bernardi and Di Lisa, 2015*). Cyclosporin A (CsA), a well-characterized mPTP inhibitor, inhibits the mPTP by binding and sequestering cyclophilin D, a mitochondrially localized peptidyl prolyl isomerase that helps catalyze pore formation (*Basso et al., 2005*; *Nakagawa et al., 2005*). Genetic inhibition of cyclophilin D protects against mPTP formation (*Baines et al., 2005*), and CsA has been shown to extend lifespan in *Caenorhabditis elegans* (*Ye et al., 2014*). Thus, the mPTP appears to be an important modulator of healthspan and lifespan.

The identification of the proteins that make up the mPTP is controversial. Recent models have moved away from the models that invoked the voltage-dependent anion channel (VDAC), mitochondrial phosphate carrier (PiC), and translocator protein (TPSO) due to genetic ablation studies showing that the mPTP can still occur in their absence (*Kwong and Molkentin, 2015*). Due to a multitude of recent studies, many experts in the field of mPTP have pointed to the F-ATP synthase (complex V) as the most probable inner mitochondrial pore candidate (*Bonora et al., 2017*; *Carraro et al., 2019*; *Mnatsakanyan and Jonas, 2020*). F-ATP synthase is able to bind cyclophilin D and form $Ca^{2+}$

currents (*Bernardi and Di Lisa, 2015*; *Mnatsakanyan and Jonas, 2020*). Some models posit that dimeric forms of F-ATP synthase open to form a pore while other models have suggested that the pore occurs via the membrane-bound proton-driving c-ring rotor (*Alavian et al., 2014*; *Azarashvili et al., 2014*; *Bonora et al., 2013*; *Bonora et al., 2017*; *Giorgio et al., 2013*; *Mnatsakanyan et al., 2019*; *Neginskaya et al., 2019*; *Urbani et al., 2019*). Despite mounting evidence supporting F-ATP synthase as the pore-forming component of the mPTP, systematic deletion of nearly every subunit of F-ATP synthase in a cell model showed that a pore is still capable of forming (*Carroll et al., 2019*; *He et al., 2017a*; *He et al., 2017b*), leading some groups to suggest that in the absence of an intact F-ATP synthase, smaller low-conductance CsA-dependent pores distinct from the mPTP form (*Neginskaya et al., 2019*). Other groups have proposed that the mPTP can be mediated by adenine nucleotide transporter (ANT), which exchanges ADP and ATP across the IMM. Genetic inhibition of ANTs helps prevent pore formation (*Karch et al., 2019*), and the 'multi-pore model' posits that the mPTP can be mediated by ANT, as well as a cyclophilin D binding structure, such as F-ATP synthase, which would explain why deletion of putative pore components may still yield pore formation (*Carraro et al., 2019*; *Karch et al., 2019*; *Carrer et al., 2021*).

ATP5O, also known as oligomycin sensitivity-conferring protein (OSCP), a subunit of the F-ATP synthase that regulates ATPase rotational activity to provide efficient ATP production (*Murphy et al., 2019*), has emerged as an important regulator of the mPTP. OSCP confers protection against the mPTP under low pH conditions and loss of OSCP increases propensity for mPTP formation *in vitro* (*Antoniel et al., 2018*; *Giorgio et al., 2013*). In mice, levels of OSCP decrease with normal aging (*Gauba et al., 2017*). In mouse models of AD, OSCP binds to amyloid beta (Aβ) and the propensity for mPTP formation increases, suggesting that the destabilization of OSCP contributes to mPTP formation (*Beck et al., 2016b*). Conversely, OSCP overexpression protects from mPTP initiation in AD and cardiac dysfunction models (*Beck et al., 2016b*; *Guo et al., 2020*). Thus, OSCP appears to be an important regulator of aging and disease progression, possibly via its ability to modulate mPTP formation.

Under mitochondrial stress, the mitochondria attempt to repair the damage, recycle damaged mitochondria, or, under deleterious circumstances, initiate cell death. Similar to the endoplasmic reticulum unfolded protein response (UPR$^{ER}$) and the cytoplasmic heat shock response (HSR), the mitochondrial unfolded protein response (UPR$^{mt}$) is capable of initiating a broad-range transcriptional response that, among other functions, aids in the refolding of mitochondrial matrix proteins (*Naresh and Haynes, 2019*). Recent studies also show that a loss of mitochondrial membrane potential (MMP) correlates with activation of the UPR$^{mt}$, and disruption of mitochondrial processes other than protein misfolding, such as those involved in TCA cycle and lipid catabolism, also induce the UPR$^{mt}$ (*Rolland et al., 2019*). UPR$^{mt}$ activation is associated with longevity and improvement in neurodegenerative models (*Durieux et al., 2011*; *Houtkooper et al., 2013*; *Kim et al., 2016*; *Merkwirth et al., 2016*; *Sorrentino et al., 2017*; *Tian et al., 2016*), but it has also conversely been shown to increase neurodegeneration, propagate mtDNA mutations, and exacerbate ischemic conditions (*Lin et al., 2016*; *Martinez et al., 2017*; *Yung et al., 2019*), underscoring its complexity. If left unmitigated, UPRs can initiate cell death (*Iurlaro and Muñoz-Pinedo, 2016*; *Münch and Harper, 2016*). Thus, the context or cellular environment are important determinants of whether UPR$^{mt}$ induction results in beneficial or detrimental effects.

In *C. elegans*, mild mitochondrial perturbations early in life can extend lifespan. Loss of OSCP/*atp-3* has previously been shown to extend lifespan when initiated during larval development (*Dillin et al., 2002*; *Rea et al., 2007*). In contrast, here, we have determined that loss of OSCP/*atp-3* during adulthood leads to initiation of the mPTP, the UPR$^{mt}$, and a shortened lifespan. Surprisingly, *atfs-1*, the UPR$^{mt}$ master transcription factor (*Haynes et al., 2010*; *Nargund et al., 2012*), helps drive the reduction of lifespan, suggesting that the UPR$^{mt}$ program can promote aging during adulthood. The adult UPR$^{mt}$ is responsive to mPTP regulators, including the immunosuppressive drug, CsA, as well as a mitochondrially localized cyclophilin and ANTs, pointing to a previously undiscovered coordination between the UPR$^{mt}$ and the mPTP. We find that the proton-driving rotor subunit as well as subunits important for dimerization of the F-ATP synthase are essential for transducing the adult UPR$^{mt}$. Loss of these subunits as well as pharmacological CsA treatment restores lifespan due to loss of OSCP/*atp-3*. These results are consistent with current models that posit that the F-ATP synthase forms the mPTP (*Bernardi and Di Lisa, 2015*). Overall, our findings point to a model in which loss of OSCP/*atp-3* in adults induces mPTP formation with subsequent activation of the UPR$^{mt}$.

Understanding the relationship between these two mitochondrial processes will further our understanding of aging as well as disparate age-related disorders, including neurodegenerative diseases, cancer, heart attack, and stroke.

## Results

### Loss of OSCP/*atp-3* during adulthood induces mPTP characteristics

The opening of the mPTP is characterized by a loss of MMP as well as an increase in cytosolic $Ca^{2+}$ and responsiveness to the mPTP inhibitor, CsA. We observed that a reduction in the abundance of OSCP/*atp-3* by RNA interference (RNAi) during adulthood caused a loss of MMP as measured by the mitochondrial dye, tetramethylrhodamine methyl ester (TMRM), while RNAi of other OXPHOS subunits from complex I, IV, and V had no effect on the MMP (*Figure 1A, B*). RNAi of OSCP/*atp-3* during adulthood also caused an increase in cytosolic $Ca^{2+}$ as measured by the intestinal FRET-based $Ca^{2+}$ reporter, KWN190, while RNAi of subunits from complex IV and complex V did not (*Figure 1D, E*). CsA rescued the loss of MMP and suppressed the rise of cytosolic $Ca^{2+}$ caused by RNAi of OSCP/*atp-3* during adulthood (*Figure 1C, F*). Loss of OSCP/*atp-3* also induced mitochondrial swelling and fragmentation compared to control, which was rescued by CsA (*Figure 1G–J*). To determine if a loss of MMP during adulthood was sufficient to recapitulate mPTP characteristics, we tested the effects of FCCP, a potent mitochondrial uncoupler. FCCP induced a loss of MMP when administered during adulthood, but did not lead to an increase in cytosolic $Ca^{2+}$ (*Figure 1—figure supplement 1A, B*), suggesting that a loss in MMP is not sufficient to recapitulate mPTP characteristics. Similarly, loss of OSCP/*atp-3* RNAi during development, which leads to a loss of MMP, did not lead to an increase in cytosolic $Ca^{2+}$ (*Figure 1—figure supplement 1C, D*), demonstrating that phenotypes that result from a loss of OSCP/*atp-3* are distinct during adulthood versus development. Overall, these results suggest that loss of OSCP/*atp-3* during adulthood uniquely induces the mPTP in *C. elegans*.

To determine if RNA expression levels of OSCP/*atp-3* may be higher during adulthood compared to other OXPHOS subunits, which would sensitize it to RNAi, we examined available RNAseq data from wormbase.org. Fragments per kilobase of transcript per million (FPKM) expression values of various OXPHOS subunits collected at young adulthood (YA) showed that OSCP/*atp-3* did not display higher expression compared to other subunits, and that its expression during larval stages (L1) and YA was also comparable (*Figure 1—figure supplement 1G*). To verify that the lack of a phenotypes from the other OXPHOS subunits was not due to inefficient RNAi, we checked for RNAi efficiency via qPCR. We observed efficient mRNA reduction for all tested subunits, suggesting that the RNAi was effective (*Figure 1—figure supplement 1E*). We also examined protein levels for subunits in which antibodies were available and observed that RNAi of the complex I subunit NUO-2 and complex V subunits ATP-1 and ATP-2 resulted in significant knockdown of protein levels (*Figure 1—figure supplement 1F*, *Figure 1—figure supplement 1—source data 1–3*). These findings support our conclusions that loss of OSCP/*atp-3* uniquely recapitulates mPTP characteristics during adulthood.

### Loss of OSCP/*atp-3* during adulthood induces a unique UPR^mt

A recent study showed that a loss of MMP in *C. elegans* during development is associated with induction of the UPR^mt (*Rolland et al., 2019*). To determine if RNAi of OSCP/*atp-3* selectively induces the UPR^mt during adulthood due to its observed loss in MMP (*Figure 1A, B*), we utilized a GFP reporter under the promoter of the UPR^mt chaperone, *hsp-6* (p*hsp-6*::GFP) (*Yoneda et al., 2004*), and compared it to select representative OXPHOS genes encoding complex I, III, IV, and V subunits. RNAi of OXPHOS subunits induced little to no UPR^mt if initiated after the last larval stage (L4), which we termed the post-developmental UPR^mt (pdvUPR^mt) (*Figure 2A,B*, *Figure 2—source data 1*). The exception was RNAi of OSCP/*atp-3*, which induced a robust UPR^mt in young adults (*Figure 2A, B*, *Figure 2—source data 1*), the timing of which corresponded with the loss in MMP (*Figure 1A, B*). In contrast, RNAi of all the same genes induced a robust UPR^mt if initiated during the early larval stages (L1, L2, and L3) of development (dvUPR^mt) (*Figure 2A, B*). These results are consistent with previous reports demonstrating that the UPR^mt is robustly induced during development but poorly induced during adulthood in *C. elegans* (*Durieux et al., 2011*; *Labbadia and Morimoto, 2015*). Treating

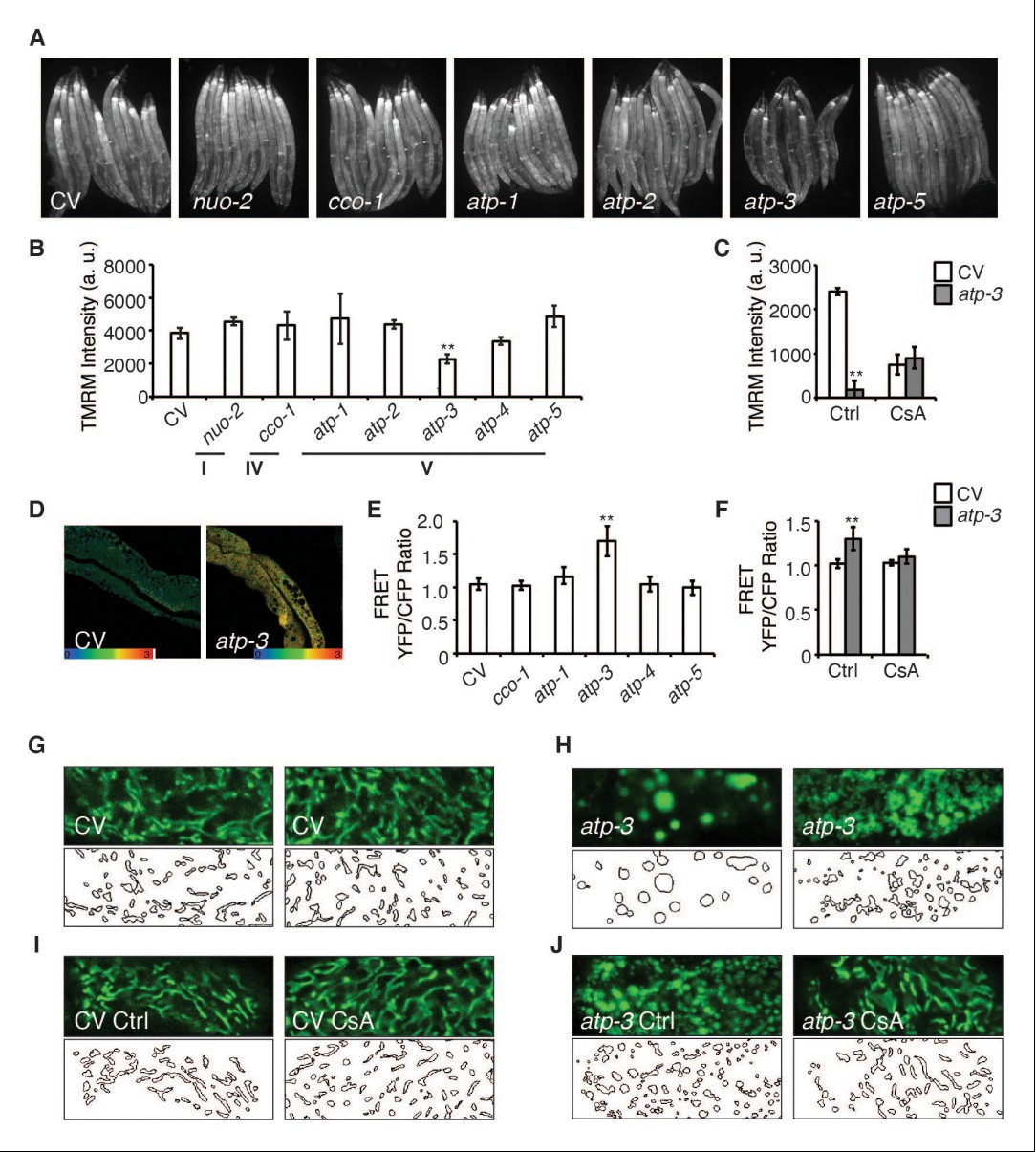

**Figure 1.** Loss of OSCP/*atp-3* during adulthood selectively recapitulates mitochondrial permeability transition pore (mPTP)-like characteristics. (**A**) Photomicrographs of mitochondrial membrane potential (MMP) as measured by tetramethylrhodamine methyl ester (TMRM) after RNA interference (RNAi) of OXPHOS subunits. RNAi and TMRM were administered for 48 hr beginning at young adulthood. Representative micrographs shown. (**B**) Quantification of TMRM intensity from (**A**). Data are the mean ± SEM of ≤ 15 animals combined from three biological experiments. **p≤0.01 by Student's *t*-test. I, IV, and V correspond to OXPHOS complexes. a.u.: arbitrary units. (**C**) Quantification of MMP after RNAi of OSCP/*atp-3* and treatment with cyclosporin A (CsA: 15 μM). RNAi and CsA were administered beginning at young adulthood for 24 hr. Data are the mean ± SEM of ≤ 15 animals combined from three biological experiments. *p≤0.05, **p≤0.01 by Student's *t*-test. (**D**) Confocal micrograph of intestinal cytosolic $Ca^{2+}$ as measured by the FRET-based calcium indicator protein D3cpv/cameleon after OSCP/*atp-3* RNAi. RNAi was administered for 48 hr beginning at young adulthood. Representative micrograph shown. (**E**) Quantification of FRET YFP/CFP ratio after RNAi of OXPHOS subunits. RNAi was administered for 48 hr beginning at young adulthood. Data are the mean ± SEM of ≤ 15 animals combined from three biological experiments. **p≤0.01 by Student's *t*-test. (**F**) Quantification of FRET YFP/CFP ratio after RNAi of OSCP/*atp-3* and treatment with CsA (15 μM). RNAi and CsA were administered beginning at young adulthood for 48 hr. Data are the mean ± SEM of ≤ 15 animals combined from three biological experiments. **p≤0.01 by Student's *t*-test. (**G–J**) Confocal micrographs of intestinal mitochondria labeled with GFP (p*ges-1*::GFP^mt) in young adults. RNAi and CsA were administered for 48 hr beginning at young adulthood, then worms were removed from the RNAi and CsA

*Figure 1 continued on next page*

*Figure 1 continued*

and aged until day 7 of adulthood followed by collection for microscopy. Top panels: fluorescent channel; bottom panels: rendering of individual mitochondria. CV: control vector; Ctrl: solvent control; CsA: 15 µM. See Materials and methods for details on rendering.

The online version of this article includes the following source data and figure supplement(s) for figure 1:

**Figure supplement 1.** Loss of OSCP/*atp-3* RNAi during adulthood uniquely causes a loss in membrane potential and a rise in cytosolic calcium.

**Figure supplement 1—source data 1.** Source data for immunoblot (*Figure 1—figure supplement 1F*).

**Figure supplement 1—source data 2.** Source data for immunoblot (*Figure 1—figure supplement 1F*).

**Figure supplement 1—source data 3.** Source data for immunoblot (*Figure 1—figure supplement 1F*).

worms with FCCP during adulthood did not induce the UPR$^{mt}$ (*Figure 2—figure supplement 1A*), indicating that loss of MMP per se is not sufficient to induce the UPR$^{mt}$ during adulthood. Post-developmental loss of OSCP/*atp-3* increased endogenous transcript levels of *hsp-6* as well as endogenous HSP-6/mtHSP70 protein levels (*Figure 2—figure supplement 1B, C*, *Figure 2—figure supplement 1—source data 1*). Post-developmental loss of OSCP/*atp-3* mildly induced the mitochondrial chaperone reporter p*hsp-60*::GFP (*Figure 2—figure supplement 1D, E*). Neither the UPR$^{ER}$ nor the HSR were induced by post-developmental loss of OSCP/*atp-3* (*Figure 2—figure supplement 1E*). RNAi of other mitochondrial genes that are known to induce a dvUPR$^{mt}$, *clk-1* (coenzyme Q hydroxylase), *mrps-5* (mitochondrial ribosome), and *tomm-22* (translocase of outer mitochondrial membrane) (*Baker et al., 2012*; *Bennett et al., 2014*; *Houtkooper et al., 2013*), did not induce the pdvUPR$^{mt}$ (*Figure 2—source data 1*). Importantly, we found that the pdvUPR$^{mt}$ was dependent on the master UPR$^{mt}$ transcription factor, *atfs-1* (*Figure 2C*; *Haynes et al., 2010*), demonstrating that the pdvUPR$^{mt}$ is regulated similarly to the previously described dvUPR$^{mt}$. Thus, loss of OSCP/*atp-3* induces a robust and specific pdvUPR$^{mt}$, which is dependent on the conserved transcription factor ATFS-1.

## Loss of OSCP/*atp-3* during adulthood shortens lifespan

Previous reports have shown that loss of OSCP/*atp-3* initiated during development robustly increases lifespan in *C. elegans* (*Dillin et al., 2002*), but how loss of OXPHOS subunits during adulthood affects lifespan has not been well studied. We initiated OSCP/*atp-3* RNAi during both development and post-development. As previously reported, we found that continuous RNAi treatment initiated during development (beginning from eggs) led to lifespan extension (*Figure 2D*, *Figure 2—source data 2*). Worms continuously exposed to OSCP/*atp-3* RNAi during post-development experienced a high incidence of matricide (data not shown). To circumvent this outcome, we administered OSCP/*atp-3* RNAi to young adults for 48 hr of adulthood and observed approximately a 38% decrease in lifespan independent of matricide (*Figure 2D*, *Figure 2—source data 2*). Loss of other OXPHOS subunits had little or no effect on lifespan when administered during adulthood for 48 hr (*Figure 2E–H*, *Figure 2—source data 2*). Surprisingly, when putative null *atfs-1(tm4525)* mutants were exposed to OSCP/*atp-3* RNAi during adulthood for 48 hr, we observed only about a 19% decrease in lifespan, suggesting that the initiation of the pdvUPR$^{mt}$ via *atfs-1* contributes to reduced lifespan (*Figure 2I*, *Figure 2—source data 2*). Thus, we have identified that loss of OSCP/*atp-3* has distinct effects in lifespan depending on if RNAi is initiated during adulthood versus development, which has not been previously described in *C. elegans*.

To further probe the developmental versus post-developmental effects on longevity from the loss of OSCP/*atp-3*, we tested how post-developmental RNAi of OSCP/*atp-3* would affect the longevity of worms treated with COX5B/*cco-1* RNAi during development, which has been shown to be sufficient to extend lifespan (*Durieux et al., 2011*). Developmental treatment with COX5B/*cco-1* RNAi followed by post-developmental treatment with OSCP/*atp-3* RNAi did not significantly alter the long-lived lifespan (*Figure 2—figure supplement 1F*, *Figure 2—source data 2*), suggesting that the effects of developmental COX5B/*cco-1* RNAi override the post-developmental effects of OSCP/*atp-3* RNAi, potentially due to the epigenetic remodeling that occurs during development (*Merkwirth et al., 2016*; *Tian et al., 2016*; *Zhu et al., 2020*; *Shao et al., 2020*).

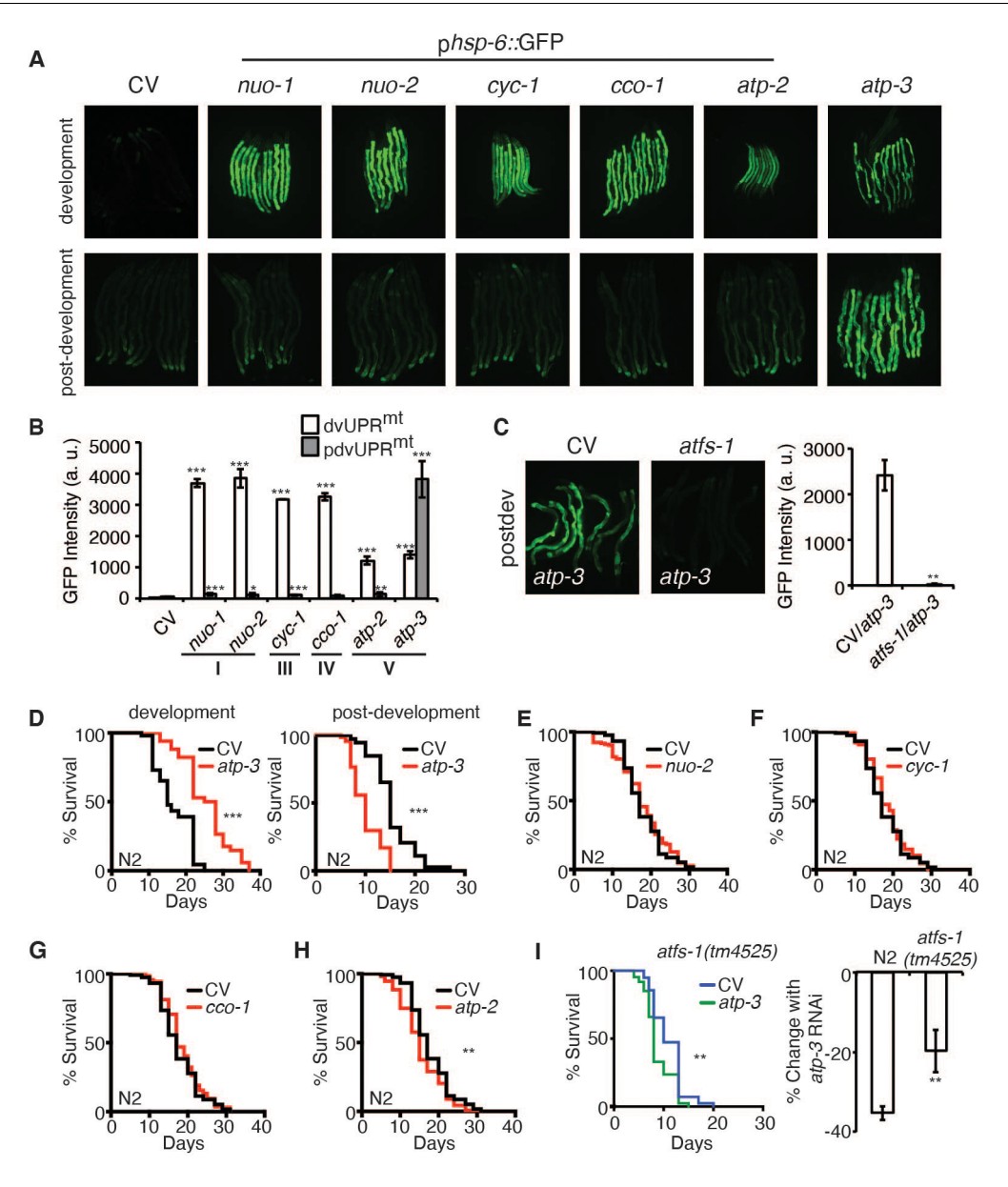

**Figure 2.** Loss of OSCP/*atp-3* during adulthood initiates a unique UPR^mt and shortens lifespan. (**A**) Photomicrographs of p*hsp-6*::GFP reporter after developmental or post-developmental RNA interference (RNAi) of OXPHOS subunits. For developmental treatment, worms were exposed to RNAi beginning from eggs for 72 hr. For post-developmental treatment, worms were exposed to RNAi beginning from young adulthood for 48 hr. CV: control vector; dev: development; post-dev: post-development. (**B**) Quantification of GFP intensity from (**A**). Data are the mean ± SEM of ≤ 15 animals combined from three biological experiments. *p≤0.05, **p≤0.01, and ***p≤0.0001 by Student's *t*-test. dv: development; pdv: post-development; I, III, IV, and V refer to OXPHOS complexes. (**C**) Photomicrographs of the p*hsp-6*::GFP reporter after RNAi of OSCP/*atp-3* and *atfs-1*. Worms were exposed to RNAi beginning from young adulthood for 48 hr. Bar graph represents quantification of GFP intensity. Data are the mean ± SEM of ≤ 15 animals combined from three biological experiments. **p≤0.01 by Student's *t*-test. CV: control vector RNAi; pdv: post-development. (**D**) Survival curves of wild-type N2 animals on CV or OSCP/*atp-3* RNAi. For developmental RNAi, worms were treated continuously since eggs. For post-developmental RNAi, worms were treated for 48 hr beginning at young adulthood. Representative curves selected from three biological experiments. ***p≤0.0001 by log rank (Mantel–Cox). (**E–H**) Survival curves of wild-type N2 animals on indicated RNAi initiated at young adulthood for 48 hr. Representative curves selected from three biological experiments. **p≤0.01 by log rank (Mantel–Cox). (**I**) Survival curves of *atfs-1(tm4525)* when OSCP/*atp-3* RNAi is initiated at

*Figure 2 continued on next page*

*Figure 2 continued*

young adulthood for 48 hr. Representative curves selected from three biological experiments. **p≤0.01 by log rank (Mantel–Cox). Bar graph is a quantification of percent change in lifespan of N2 or *atfs-1(tm4525)* mutant. **p≤0.01 by Student's *t*-test is the mean of the percent change in lifespan ± SEM from three biological experiments.

The online version of this article includes the following source data and figure supplement(s) for figure 2:

**Source data 1.** List of genes tested for whether they induce a developmental UPR$^{mt}$ (dvUPR$^{mt}$) or post-developmental UPR$^{mt}$ (pdvUPR$^{mt}$).

**Source data 2.** Summary of lifespans.

**Figure supplement 1.** Loss of the OSCP/*atp-3* during adulthood induces a specific post-developmental UPR$^{mt}$.

**Figure supplement 1—source data 1.** Source data for immunoblot (*Figure 2—figure supplement 1C*).

**Figure supplement 2.** The post-developmental UPR$^{mt}$ is temporally confined and reversible.

## The post-developmental UPR$^{mt}$ is temporally confined and reversible

Given that the UPR$^{mt}$ has not been studied during adulthood in *C. elegans*, we sought to determine the window of the UPR$^{mt}$ during adulthood. We initiated RNAi of OSCP/*atp-3* beginning at the last larval stage (L4 stage) and every few hours thereafter into adulthood. GFP expression was examined 48 hr after RNAi initiation (*Figure 2—figure supplement 2A*). We observed that the pdvUPR$^{mt}$ was initiated up to 6 hr after the L4 stage, after which RNAi of OSCP/*atp-3* no longer induced the UPR$^{mt}$. In contrast, RNAi of COX5B/*cco-1* of complex IV had no effects on the UPR$^{mt}$ at any of these stages (*Figure 2—figure supplement 2A*). For all subsequent post-development experiments, RNAi was therefore administered at the young adult stage corresponding to 4 hr after the L4 stage. Thus, the pdvUPR$^{mt}$ is confined to pre-gravid stages of adulthood, corresponding with previous reports showing a global decline in stress responses at the onset of egg-laying (*Labbadia and Morimoto, 2015*).

Previous studies have shown that developmental RNAi of COX5B/*cco-1* RNAi leads to persistent activation of the UPR$^{mt}$ into adulthood, even after removal from RNAi (*Durieux et al., 2011*). Similarly, we observed that developmental RNAi of OSCP/*atp-3* initiated a UPR$^{mt}$ that persisted into adulthood, even after removal from OSCP/*atp-3* RNAi (*Figure 2—figure supplement 2B*). In contrast, removal from post-developmental OSCP/*atp-3* RNAi treatment led to a steady decline in the GFP signal (*Figure 2—figure supplement 2B*), suggesting the activation of the pdvUPR$^{mt}$ is reversible.

## The post-developmental UPR$^{mt}$ is dependent on mPTP factors

To determine if the pdvUPR$^{mt}$ is initiated in response to the mPTP, we tested pharmacological and genetic modulators of the mPTP on induction of the pdvUPR$^{mt}$. CsA binds cyclophilins, which in the cytoplasm regulates calcineurin signaling (*Liu et al., 1991*; *Takahashi et al., 1989*), while in the mitochondria inhibits the mPTP (*Nicolli et al., 1996*). To parse out the mitochondrial versus cytoplasmic functions of CsA, we also tested the cytoplasmic-only immunosuppressive drug, FK506, which acts similarly to CsA in that it modulates calcineurin signaling in the cytoplasm (*Liu et al., 1991*). We observed that CsA strongly inhibited the pdvUPR$^{mt}$ but not dvUPR$^{mt}$ in a dose-dependent manner (*Figure 3A, C*, *Figure 3—figure supplement 1A*). In contrast, we found that FK506 had no effect on the pdvUPR$^{mt}$ (*Figure 3A, C*, *Figure 3—figure supplement 1B*), demonstrating that CsA acts in the mitochondria to suppress the pdvUPR$^{mt}$. The mPTP has been shown to be regulated by adenine nucleotide translocases (ANTs) of the inner mitochondrial membrane and loss of ANTs helps prevent the mPTP (*Karch et al., 2019*). We tested *ant-1.1*, which is ubiquitously expressed in *C. elegans*, and *ant-1.2*, which is expressed predominantly in the pharynx and intestines (*Farina et al., 2008*). RNAi of *ant-1.1* moderately suppressed the pdvUPR$^{mt}$ (*Figure 3E*) while RNAi of *ant-1.2* strongly suppressed the pdvUPR$^{mt}$, but not the dvUPR$^{mt}$ (*Figure 3B, D*). In mammals, CsA acts in the mitochondria to inhibit the mPTP by binding and sequestering cyclophilin D, a peptidyl prolyl isomerase (*Nicolli et al., 1996*). *C. elegans* contains 17 poorly defined cyclophilins, of which two are predicted to be mitochondrially localized, *cyn-1* and *cyn-17* (*Figure 3—source data 1*). RNAi of *cyn-1* did not inhibit the pdvUPR$^{mt}$ (*Figure 3F*) while *cyn-17* did (*Figure 3B, D*) suggesting that *cyn-17* may act similarly to cyclophilin D in mediating a conformation change that leads to the mPTP. RNAi of *cyn-17* did not affect the dvUPR$^{mt}$ (*Figure 3B, D*). Finally, we observed that CysA was able to

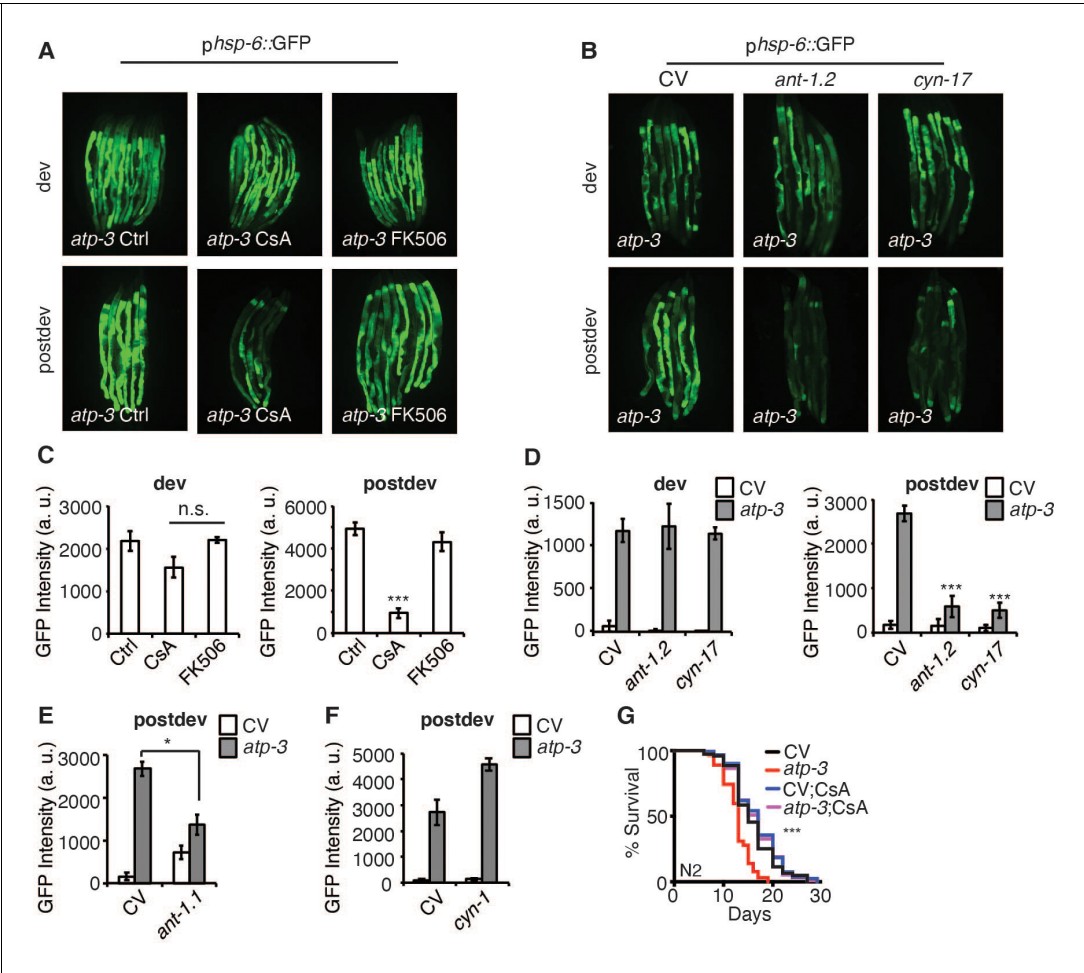

**Figure 3.** The post-developmental UPR^mt is regulated by pharmacological and genetic modulators of the mitochondrial permeability transition pore (mPTP). (A, B) Photomicrographs of p*hsp-6*::GFP reporter after developmental or post-developmental RNA interference (RNAi) and drug treatments (cyclosporin A (CsA) and FK506, 15 µM). For developmental treatment, worms were treated beginning from eggs for 72 hr. For post-developmental treatment, worms were treated beginning from young adulthood for 48 hr. dev: development; post-dev: post-development. (C, D) Quantification of GFP intensity from (A, B). Data are the mean ± SEM of ≤ 15 animals combined from three biological experiments. *p≤0.05, ***p≤0.0001 by Student's *t*-test; n.s., not significant; CV: control vector. (E, F) Quantification of GFP intensity from p*hsp-6*::GFP reporter after RNAi treatment for 48 hr beginning from young adulthood. (G) Survival curves of wild-type N2 animals on CV or OSCP/*atp-3* RNAi with either solvent control or CsA (15 µM). RNAi and CsA were administered beginning at young adulthood for 48 hr and then transferred to regular nematode growth medium (NGM) plates for the remainder of the lifespan. Representative curves selected from three biological experiments. ***p≤0.0001 by log rank (Mantel–Cox).

The online version of this article includes the following source data and figure supplement(s) for figure 3:

**Source data 1.** List of *C. elegans* cyclophilins and their predicted mitochondrial localization using the MitoFates mitochondrial targeting sequence (MTS) prediction tool.

**Figure supplement 1.** Loss of OSCP/*atp-3* during adulthood recapitulates mitochondrial permeability transition pore (mPTP) characteristics.

reverse the lifespan shortening caused by OSCP/*atp-3* RNAi (*Figure 3G*), demonstrating that inhibition of the UPR^mt can be beneficial under certain conditions. Together, these results show that the pdvUPR^mt is regulated by canonical pharmacological and genetic mPTP factors.

## Loss of F-ATP synthase ATPases induces a post-developmental UPR^mt

F-ATP synthase is composed of a membrane-bound proton-driving rotor (Fo), a catalytic ATPase that converts ADP to ATP (F1), and peripheral stalk and supernumerary subunits that help bridge these two portions together (*Figure 4E*). OSCP/*atp*-3 sits on the ATPase and helps tether it to the peripheral stalk subunits. We systematically tested via RNAi whether loss of F-ATP synthase subunits

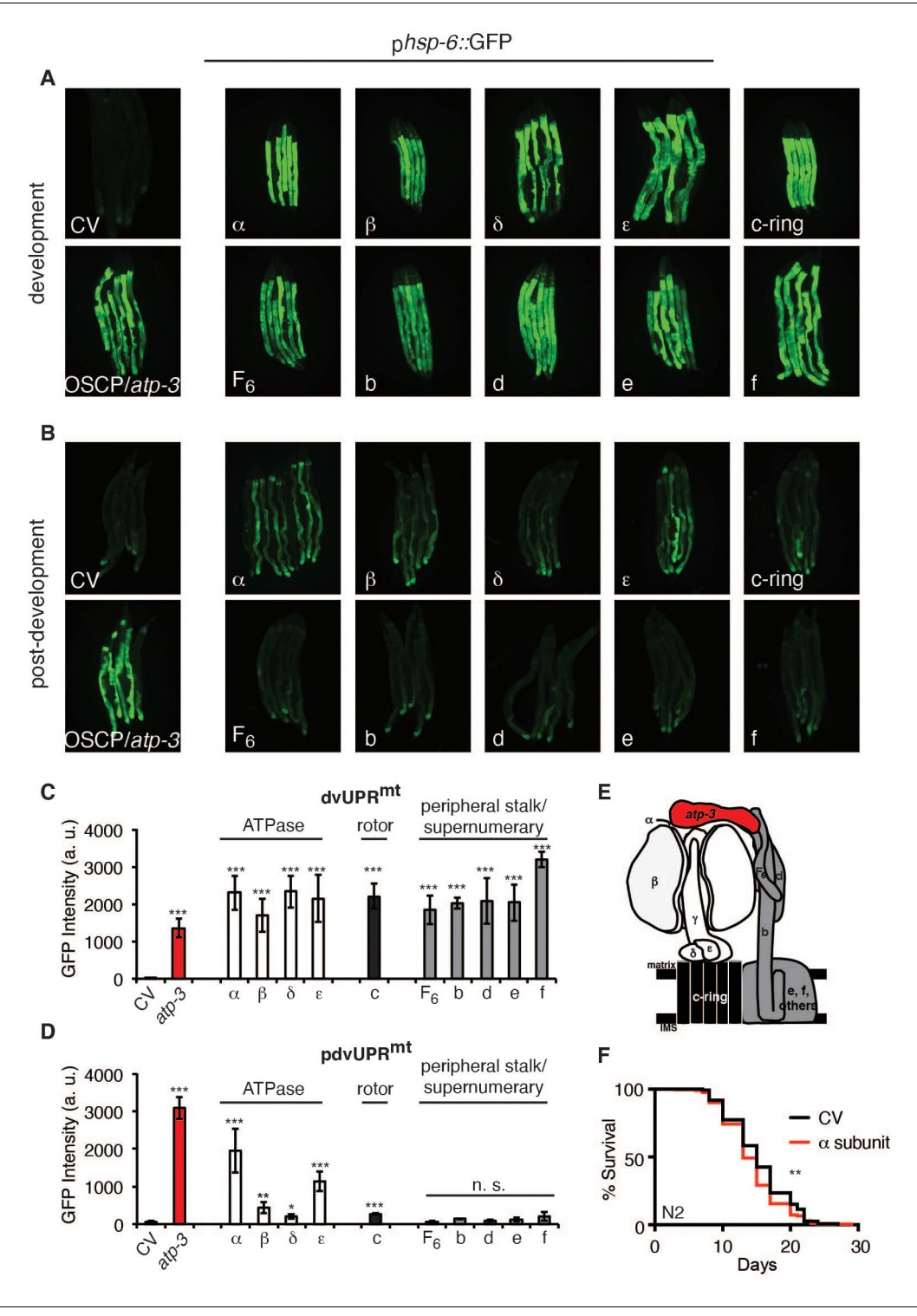

**Figure 4.** Loss of F-ATP synthase ATPases induces a post-developmental UPR^mt. (**A**) Developmental RNA interference (RNAi) of all ATP synthase subunits tested induced the p*hsp-6*::GFP reporter. For developmental treatment, worms were exposed to RNAi beginning from eggs for 72 hr. (**B**) Post-developmental RNAi of ATPase and c-ring subunits but not peripheral stalk or supernumerary subunits of the F-ATP synthase induced the p*hsp-6*:: GFP reporter. Worms were exposed to RNAi beginning from young adulthood for 48 hr. post-dev: post-development. (**C**) Quantification of GFP intensity from (**A**). Data are the mean ± SEM of ≤ 15 animals combined from three biological experiments. ***p≤0.0001 by Student's *t*-test. dv: developmental. (**D**) Quantification of GFP

*Figure 4 continued on next page*

*Figure 4 continued*
intensity from (**B**). Data are the mean ± SEM of ≤ 15 animals combined from three biological experiments. *p≤0.05, **p≤0.01, and ***p≤0.0001 by Student's *t*-test. pdv: post-developmental. (**E**) Schematic of monomeric F-ATP synthase. White subunits: ATPase; black subunits: H+-rotor/c-ring; gray subunits: peripheral stalk and supernumerary subunits; red subunit: oligomycin sensitivity-conferring protein (OSCP/*atp-3*). (**F**) Survival curves of wild-type N2 animals on CV or α/*atp-1* RNAi initiated at young adulthood for 48 hr. Pooled survival curves from four biological experiments. **p≤0.01 by log rank (Mantel–Cox). CV: control vector.

The online version of this article includes the following source data for figure 4:

**Source data 1.** Summary of the effects of RNA interference (RNAi) of F-ATP synthase subunits on the developmental UPR$^{mt}$ (dvUPR$^{mt}$) or post-developmental UPR$^{mt}$ (pdvUPR$^{mt}$).

other than OSCP/*atp-3* could induce a pdvUPR$^{mt}$. During development, loss of rotor subunits, ATPase subunits, or peripheral stalk and supernumerary subunits all induced a robust UPR$^{mt}$ (*Figure 4A, C*). In contrast, during adulthood, loss of rotor subunits (c-ring/*Y82E9BR.3*), peripheral stalk or supernumerary subunits (F6/*atp-4*, d/*atp-5*, b/*asb-2*, e/*R04F11.2*, f/*R53.4*), induced little to no UPR$^{mt}$ (*Figure 4B, D*, *Figure 4—source data 1*). Loss of the ATPase subunits (α/*atp-1*, β/*atp-2*, δ/*F58F12.1*, ε/*hpo-18*) induced a mild to moderate UPR$^{mt}$, though none as robustly as loss of OSCP/*atp-3*. We observed that post-developmental loss of the α/*atp-1* ATPase subunit exhibited the second most robust pdvUPR$^{mt}$ (*Figure 4B, D*) but did not induce a loss of MMP or a rise in cytosolic Ca$^{2+}$ (*Figure 1A–C*). Consistently, we observed a minor shortening of survival due to post-developmental RNAi of α/*atp-1* (*Figure 4F*). These findings support our hypothesis that loss of OSCP/*atp-3*, but not other F-ATP synthase subunits, activates an mPTP that is coupled to a maladaptive pdvUPR$^{mt}$.

## Loss of F-ATP synthase subunits important for the formation of the mPTP suppresses the post-developmental UPR$^{mt}$

Current models posit that the F-ATP synthase forms a pore that is capable of releasing Ca$^{2+}$ under conditions of high oxidative stress, leading to rupturing of the mitochondria and initiation of cell death cascades. Some models suggest that F-ATP synthase dimers form the mPTP (*Figure 5E*) and that peripheral and supernumerary subunits are essential for pore formation (*Carraro et al., 2014*; *Giorgio et al., 2013*; *Guo et al., 2019*; *Urbani et al., 2019*). Other models demonstrate that F-ATP synthase monomers are sufficient for the mPTP and specify the c-ring proton-driving rotor as the actual pore-forming component (*Figure 5F*; *Alavian et al., 2014*; *Azarashvili et al., 2014*; *Bonora et al., 2013*; *Bonora et al., 2017*; *Mnatsakanyan et al., 2019*; *Neginskaya et al., 2019*). To determine whether the structural integrity of F-ATP synthase subunits was required for the pdvUPR$^{mt}$, we systematically knocked down OSCP/*atp-3* as well as one additional F-ATP synthase subunit via RNAi. When we knocked down the c-ring subunits (color coded black) as well as peripheral and supernumerary subunits (color coded gray) via RNAi in adults, we observed nearly complete inhibition of the OSCP/*atp-3* RNAi-mediated pdvUPR$^{mt}$ (*Figure 5B, D*). When we knocked down the ATPase subunits (color coded white) in adults, we observed that loss of the β/*atp-2* subunit robustly suppressed the OSCP/*atp-3* RNAi-mediated pdvUPR$^{mt}$, possibly due to its role in modulating Ca$^{2+}$ in the mPTP, while loss of α/*atp-1* moderately inhibited the pdvUPR$^{mt}$ (*Figure 5B, D*). Loss of the ATPase subunits δ/*F58F12.1* or ε/*hpo-18* in adults did not affect the pdvUPR$^{mt}$ (*Figure 5B, D*). In contrast, dual loss of subunits during development all robustly activated the dvUPR$^{mt}$ (*Figure 5A, C*). Dual loss of subunits from other OXPHOS complexes (NDUFS3/*nuo-2*, complex I; COX5B/*cco-1*, complex IV) had no effect or slightly increased the dvUPR$^{mt}$ and the pdvUPR$^{mt}$ (*Figure 5—figure supplement 1A, B*). Thus, we find subunits critical for dimerization (peripheral and supernumerary subunits) and proton translocation (c-ring rotor) are required to transduce the OSCP/*atp-3* RNAi-mediated pdvUPR$^{mt}$. We also find that the β/*atp-2* subunit, previously found to play an important role in Ca$^{2+}$ mediated mPTP (*Giorgio et al., 2017*), is required to transduce the OSCP/*atp-3* RNAi-mediated pdvUPRmt. Taken together, these findings support a model in which inhibition of the mPTP via deletion of critical F-ATP synthase subunits inhibits the pdvUPR$^{mt}$.

To verify that the use of dual-RNAi did not interfere with knockdown of OSCP/*atp-3*, we assessed an endogenously expressing p*atp-3*::ATP-3::GFP translational reporter generated via CRISPR-Cas-9 (*Figure 5—figure supplement 1C*). We observed efficient knockdown of ATP-3 via RNAi in the

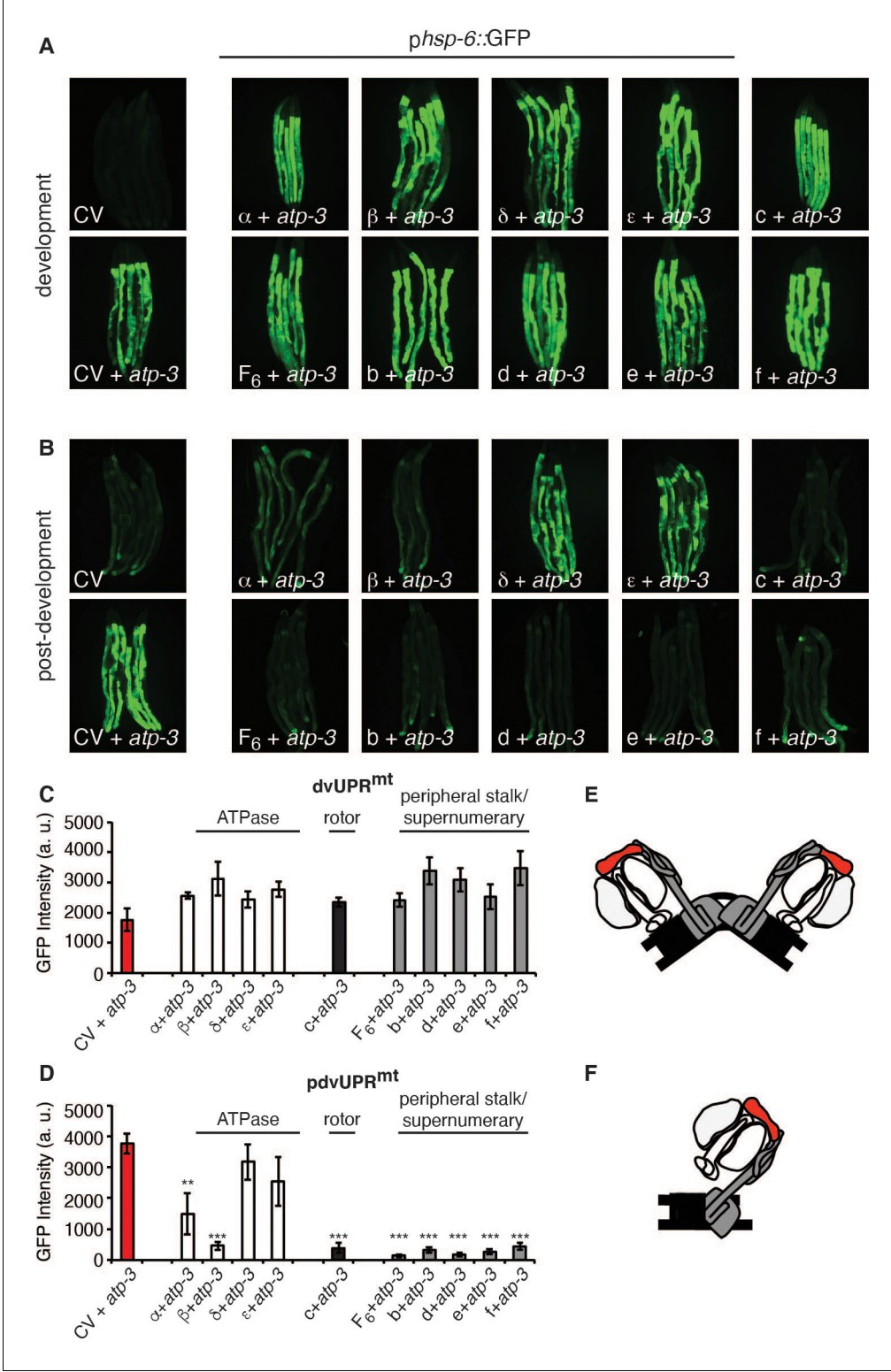

**Figure 5.** Loss of F-ATP synthase subunits important for the formation of the mitochondrial permeability transition pore (mPTP) suppresses the post-developmental UPR[mt]. (**A, B**) Concomitant RNA interference (RNAi) of OSCP/ *atp-3* and individual F-ATP synthase subunits during developmental (**A**) or post-developmental (**B**) modulated the UPR[mt] to varying degrees in the p*hsp-6*::GFP reporter strain. (**C, D**) Quantification of GFP intensity from (**A, B**). Data are the mean ± SEM of ≤ 15 animals combined from three biological experiments. **p≤0.001, ***p≤0.0001 compared to control vector (CV) + *atp-3* condition by Student's *t*-test. dv: developmental; pdv: post-

*Figure 5 continued*

developmental. (**E**, **F**) Models of F-ATP synthase forming a dimeric mPTP (**E**) or monomeric mPTP (**F**). White subunits: ATPase; black subunits: H+ rotor/c-ring; gray subunits: peripheral stalk and supernumerary subunits; red subunit: oligomycin sensitivity-conferring protein (OSCP/*atp-3*).

The online version of this article includes the following source data and figure supplement(s) for figure 5:

**Figure supplement 1.** Effects of loss of OXPHOS subunits on the dvUPR^mt and pdvUPR^mt.

**Figure supplement 1—source data 1.** Source data for immunoblot (*Figure 5—figure supplement 1D*).

presence of either β/*atp-2* or d/*atp-5* RNAi, two subunits that suppress the pdvUPR^mt. These findings demonstrate that inhibition of the pdvUPR^mt is not due to ineffective RNAi of OSCP/*atp-3* but rather the functional consequence of removing additional F-ATP synthase subunits. To examine the effects of dual-RNAi another way, we examined how loss of F-ATP synthase subunits impacted the pdvUPR^mt after α/*atp-1* RNAi, which induced the second most robust pdvUPR^mt (*Figure 4B, D*). Remarkably, we see the same pattern of pdvUPR^mt activation and inhibition as with loss of OSCP/*atp-3*: loss of the ATPase subunit β/*atp-2* and peripheral stalk subunit d/*atp-5* suppressed the α/*atp-1* RNAi-mediated pdvUPR^mt while loss of NDUFS3/*nuo-2* and COX5B/*cco-1* had no effect (*Figure 5—figure supplement 1D*). Importantly, immunoblots against α/*atp-1* showed similar protein knockdown under all conditions (*Figure 5—figure supplement 1—source data 1*), confirming that dual-RNAi is an effective method to assess the structural components of the F-ATP synthase.

## Loss of F-ATP synthase subunits important for the formation of the mPTP reverses mPTP characteristics and regulates longevity

Based on our observations that loss of peripheral stalk subunits is capable of suppressing the pdvUPR^mt, we tested if their loss would also suppress mPTP characteristics. RNAi of peripheral stalk subunits (F6/*atp-4*, d/*atp-5*) or the proton-driving rotor c-ring/*Y82E9BR.3* rescued the loss in MMP, suppressed the rise in cytosolic Ca^{2+}, and rescued the shortened lifespan caused by RNAi of OSCP/*atp-3* (*Figure 6A–G*, *Figure 2—source data 2*). Interestingly, RNAi of b/*asb-2* did not rescue the loss in membrane potential but did inhibit the rise in cytosolic Ca^{2+} and rescued lifespan (*Figure 6A–D*, *Figure 2—source data 2*). We further examined the intestinal mitochondrial morphology in worms dually treated with OSCP/*atp-3* and either F6/*atp-4*, d/*atp-5*, or c-ring/*Y82E9BR.3* RNAi. Though RNAi of the peripheral and rotor subunits on its own caused aberrant mitochondrial morphology distinct from controls, RNAi treatment rescued the rounded and swollen mitochondrial morphology observed due to OSCP/*atp-3* RNAi (*Figure 6I–L*). Thus, inhibition of key subunits of F-ATP synthase generally reverses the detrimental effects associated with the mPTP/pdvUPR^mt nexus.

While testing the epistatic relationship of the F-ATP synthase subunits, we observed that post-developmental RNAi of c-ring/*Y82E9BR.3* on its own was sufficient to significantly extend lifespan (*Figure 6H*), while the loss of the b/*asb-2*, F6/*atp-4*, and d/*atp-5* peripheral stalk subunits did not (*Figure 6—figure supplement 1*, *Figure 2—source data 2*). Similarly, we had previously observed that loss of the α/*atp-1* and β/*atp-2* ATPase subunits had minor lifespan shortening effects (*Figures 2H* and *3F*). Thus, loss of the proton-driving c-ring during adulthood uniquely extends lifespan, which is particularly intriguing due to the fact that a plethora of evidence supports a role for the c-ring as the pore-forming component of mPTP (*Alavian et al., 2014*; *Azarashvili et al., 2014*; *Bonora et al., 2013*; *Bonora et al., 2017*; *Mnatsakanyan et al., 2019*; *Neginskaya et al., 2019*) and its involvement in disease (*Amodeo et al., 2021*; *Licznerski et al., 2020*; *Morciano et al., 2021*).

## Discussion

While loss of the F-ATP synthase subunit OSCP/*atp-3* during development leads to lasting activation of the UPR^mt and is associated with longevity, we have discovered that loss of this subunit during adulthood induces the mPTP and activates a reversible and *atfs-1*-dependent UPR^mt (pdvUPR^mt). Furthermore, we observed that activation of the pdvUPR^mt helps drive aging. Suppression of the mPTP/UPR^mt via genetic or pharmacological interventions is protective. Loss of other OXPHOS

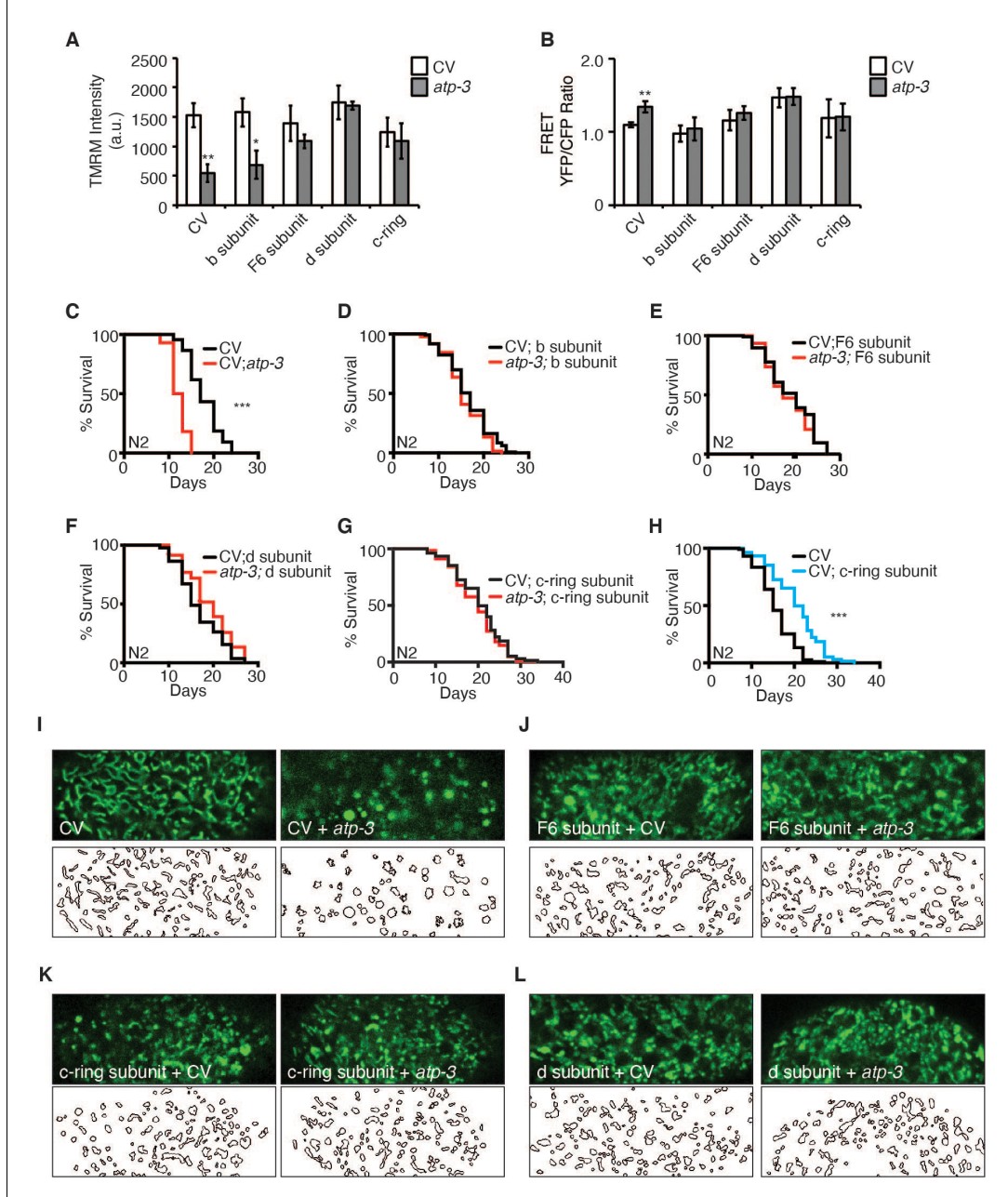

**Figure 6.** Reversal of mitochondrial permeability transition pore (mPTP) characteristics and regulation of longevity by F-ATP synthase subunits. (**A**) Testing for epistatic interactions between OSCP/*atp-3* and F-ATP synthase subunits on the mitochondrial membrane potential (MMP). RNA interferences (RNAis) were concomitantly administered beginning at young adulthood for 48 hr. Tetramethylrhodamine methyl ester (TMRM) was spotted on seeded plates. Data are the mean ± SEM of ≤ 15 animals combined from four biological experiments. *p≤0.05, **p≤0.01 by Student's *t*-test. CV: control vector. (**B**) Testing for epistatic interactions between OSCP/*atp-3* and F-ATP synthase subunits on cytosolic Ca²⁺ using the FRET-based calcium indicator protein D3cpv/cameleon. RNAis were concomitantly administered beginning at young adulthood for 48 hr. Data are the mean ± SEM of ≤ 15 animals combined from three biological experiments. **p≤0.01 by Student's *t*-test. (**C–G**) Testing for epistatic interactions between OSCP/*atp-3* and F-ATP synthase subunits on the survival. Survival curves of wild-type N2 animals treated with RNAi beginning at young adulthood for 48 hr and then transferred to regular nematode growth medium (NGM) plates for the remainder of the lifespan. Lifespan curves from two pooled biological replicates. ***p≤0.0001 by log rank (Mantel–Cox). (**H**) Survival curves of wild-type N2 animals on CV or CV/c-ring subunit. Lifespan curves from two pooled biological replicates. ***p≤0.0001 by log rank (Mantel–Cox). CV: control vector RNAi. (**I–L**) Testing for epistatic interactions between OSCP/*atp-3* and F-ATP synthase subunits on mitochondrial morphology. Confocal micrographs of intestinal mitochondria labeled with GFP (p*ges-1*::GFP^mt) treated with RNAi for 48 hr beginning at young adulthood. Worms were then removed from the RNAi and aged until day 7 of adulthood followed by

*Figure 6 continued on next page*

*Figure 6 continued*

collection for microscopy. Top panels: fluorescent channel; bottom panels: rendering of individual mitochondria. See Materials and methods for details on rendering.

The online version of this article includes the following figure supplement(s) for figure 6:

**Figure supplement 1.** Post-developmental loss of F-ATP synthase peripheral stalk subunits.

subunits or administration of FCCP during adulthood does not cause a loss of the MMP, a rise in cytosolic $Ca^{2+}$, or activation of the pdvUPR$^{mt}$, suggesting that the activation of mPTP/UPR$^{mt}$ is specifically due to loss of OSCP/*atp-3*. In contrast, it does not appear that the loss of OSCP/*atp-3* during development activates an mPTP. Thus, loss of OSCP/*atp-3* can have drastically different effects depending on the life stage or cellular milieu of the organism.

While the mPTP's detrimental effects on health are well established, considerable evidence suggests that the UPR$^{mt}$ contributes to health and longevity (*Houtkooper et al., 2013*; *Merkwirth et al., 2016*; *Mouchiroud et al., 2013*; *Sorrentino et al., 2017*). Activating the UPR$^{mt}$ in neurons can activate protective cell non-autonomous signals and also epigenetically rewire *C. elegans* to live longer (*Durieux et al., 2011*; *Merkwirth et al., 2016*; *Tian et al., 2016*; *Zhang et al., 2018*; *Zhu et al., 2020*; *Shao et al., 2020*). NAD$^+$ boosters activate a UPR$^{mt}$ that contributes to longevity and ameliorates AD (*Mouchiroud et al., 2013*; *Sorrentino et al., 2017*). However, an unbiased screen that identified activators of the UPR$^{mt}$ found no correlation between UPR$^{mt}$ activation and longevity (*Bennett et al., 2014*) and constitutive activation of the UPR$^{mt}$ in dopaminergic neurons led to increased neurodegeneration (*Martinez et al., 2017*). Activation of the UPR$^{mt}$ in our setting is distinct in that it is specifically linked to activation of the mPTP. Thus, it is possible that preemptively boosting the UPR$^{mt}$ may ward off aging and disease while activation during a diseased setting may exacerbate conditions, akin to instances of inflammation in disease.

We propose a model in which loss of OSCP/*atp-3* induces a conformational change in F-ATP synthase that leads to pore formation and activation of the UPR$^{mt}$ during adulthood but not during development. Previous reports have shown that loss of OSCP increases susceptibility to $Ca^{2+}$-induced mPTP formation and that key residues within the OSCP are required to suppress the mPTP during conditions of low pH (*Antoniel et al., 2018*; *Giorgio et al., 2013*), suggesting that a functional and intact OSCP protects against pore formation. OSCP levels have also been shown to decrease with age while concomitantly increasing its binding to amyloid β, suggesting that loss of OSCP destabilizes the remaining F-ATP synthase to increase pore formation (*Beck et al., 2016b*). However, it has also been shown that OSCP provides the binding site for cyclophilin D and thus it has been proposed to be critical in the formation of the mPTP (*Giorgio et al., 2009*; *Giorgio et al., 2013*). However, immunoprecipitation studies show that cyclophilin D my also bind the peripheral stalk subunit b, which is also supported by EM studies (*Daum et al., 2013*; *Giorgio et al., 2013*). Thus, our findings support a model in which loss of OSCP/*atp-3* induces a conformation change that is favorable for cyclophilin binding to the remaining peripheral stalk proteins or to sites independent of the F-ATP synthase, leading to destabilization of F-ATP synthase, pore formation with a loss of MMP, and subsequent activation of the pdvUPR$^{mt}$.

The loss of OSCP/*atp-3* may also be more impactful than other subunits. Unlike other F-ATP synthase subunits, OSCP/*atp-3* is prominently accessible in the mitochondrial matrix and has a diverse set of binding partners, such as estradiol and p53, which can modulate ATP production (*Bergeaud et al., 2013*; *Zheng and Ramirez, 1999*). Its loss may induce a strong protein misfolding cascade, reminiscent of the He and Lemasters mPTP model first proposed in 2002 (*He and Lemasters, 2002*). In this model, it was proposed that exposure to activators of the mPTP, such as oxidants, causes protein misfolding of integral membrane proteins, thereby recruiting chaperones such as cyclophilin D for repair. If the protein misfolding is unable to be repaired, then $Ca^{2+}$, along with cyclophilin D, could catalyze pore formation in a CsA-dependent manner. Indeed, exposure to the oxidants paraquat and manganese have been shown to induce the UPR$^{mt}$ and the mPTP in separate studies (*Angeli et al., 2014*; *Costantini et al., 1995*; *Nargund et al., 2012*; *Rao and Norenberg, 2004*). However, more research is needed to clarify the relationship between protein misfolding and the mPTP.

AD and PD both display evidence of the mPTP, and in some instances, elevated UPR[mt] profiles have also been observed (*Beck et al., 2016a*; *Beck et al., 2016b*; *Ludtmann et al., 2018*; *Pérez et al., 2018*; *Sorrentino et al., 2017*), but the relationship between these two mitochondrial processes has not been fully explored. Ischemic reperfusion injuries directly cause the mPTP but little is known about the UPR[mt] under these conditions (*Kaufman and Crowder, 2015*). Establishing a clearer understanding of the relationship between the UPR[mt] and the mPTP in these disease states could result in the development of new therapeutics for these and related disorders.

# Materials and methods

## Key resources table

| Reagent type (species) or resource | Designation | Source or reference | Identifiers | Additional information |
|---|---|---|---|---|
| Strain, strain background (*Caenorhabditis elegans*) | N2 Bristol | Caenorhabditis Genetics Center | Wild-type | |
| Strain, strain background (*Caenorhabditis elegans*) | SJ4100 | Caenorhabditis Genetics Center | *hsp-6*p::GFP | |
| Strain, strain background (*Caenorhabditis elegans*) | SJ4058 | Caenorhabditis Genetics Center | *hsp-60*p::GFP | |
| Strain, strain background (*Caenorhabditis elegans*) | SJ4005 | Caenorhabditis Genetics Center | *hsp-4*p::GFP | |
| Strain, strain background (*Caenorhabditis elegans*) | CL2070 | Caenorhabditis Genetics Center | *hsp-16.2*p::GFP | |
| Strain, strain background (*Caenorhabditis elegans*) | KWN190 | Caenorhabditis Genetics Center | rnyEx109[*nhx-2*p::D3cpv + *pha-1*(+)], *pha-1(e2123)* III; (*him-5(e1490)*)V | |
| Strain, strain background (*Caenorhabditis elegans*) | ZC376.7 | National BioResource Project | *atfs-1(tm4525)* | |
| Strain, strain background (*Caenorhabditis elegans*) | PHX1826 | SunyBiotech | qIs48[*atp-3(syb1826)*]/ hT2[*bli-4(e937) let-?(q782)*] | |
| Strain, strain background (*Caenorhabditis elegans*) | SJ4143 | Caenorhabditis Genetics Center | *ges-1*p::GFP(mt) | |
| Strain, strain background (*Caenorhabditis elegans*) | GL347 | This study | SJ4100 backcrossed 6× to N2 Bristol | |
| Antibody | Anti-GRP 75 (D9) (mouse monoclonal) | Santa Cruz Biotechnology | sc-133137 | WB(1:1000) |
| Antibody | Anti-β tubulin (D-10) (mouse monoclonal) | Santa Cruz Biotechnology | sc-5274 | WB(1:1000) |

*Continued on next page*

*Continued*

| Reagent type (species) or resource | Designation | Source or reference | Identifiers | Additional information |
|---|---|---|---|---|
| Antibody | Anti-β-actin (8H10D10) (mouse monoclonal) | Cell Signaling Technology | Cat# 3700 | WB(1:1000) |
| Antibody | Anti-ATP5A1 (15H4C4) (mouse monoclonal) | Thermo Fisher | Catalog # 43-9800 | WB(1:1000) |
| Antibody | Anti-ATP synthase beta (3D5AB1) (mouse monoclonal) | Thermo Fisher | Catalog # A-21351 | WB(1:1000) |
| Sequence-based reagent | Anti-NDUFS3 (17D95) (mouse monoclonal) | Thermo Fisher | Catalog # 43-9200 | WB(1:1000) |
| Chemical compound, drug | FCCP (trifluoromethoxy carbonylcyanide phenylhydrazone) | Cayman Chemical | Item # 15218 | |
| Chemical compound, drug | FK506 (tacrolimus) | Cayman Chemical | Item # 10007965 | |
| Chemical compound, drug | Tetramethylrhodamine methyl ester (perchlorate) (TMRM) | Cayman Chemical | Item # 21437 | |
| Chemical compound, drug | Cyclosporin A | Cayman Chemical | Item # 12088 | |
| Software, algorithm | GraphPad Prism | GraphPad Software | v.9 | |
| Software, algorithm | ImageJ software | ImageJ http://imagej.nih.gov/ij/ | 1.52A | |
| Other | MitoFates tool | http://mitf.cbrc.jp/MitoFates/cgi-bin/top.cgi (*Fukasawa et al., 2015*). | | |
| Sequence-based reagent | *act-1*, forward | This study | qPCR primers | ACGACGAGTCCGGCCCATCC |
| Sequence-based reagent | *act-1*, reverse | This study | qPCR primers | GAAAGCTGGTGGTGACGATGGTT |
| Sequence-based reagent | *atp-2*, forward | This study | qPCR primers | GAAGGACAAATCTCCCCACA |
| Sequence-based reagent | *atp-2*, reverse | This study | qPCR primers | CGCCACATTCTTCCTTTTTC |
| Sequence-based reagent | *atp-4*, forward | This study | qPCR primers | AATATGTTGCCTCCCGTGAT |
| Sequence-based reagent | *atp-4*, reverse | This study | qPCR primers | GGAACAAAAACGTTCATTCG |
| Sequence-based reagent | *atp-5*, forward | This study | qPCR primers | TCTTCGACGTGCCGACAA |
| Sequence-based reagent | *atp-5*, reverse | This study | qPCR primers | AAATGGTAGGAGAGCGATAAGG |
| Sequence-based reagent | *nuo-2*, forward | This study | qPCR primers | TGAAGTTGCTGAGCCAACAC |
| Sequence-based reagent | *nuo-2*, reverse | This study | qPCR primers | TCCACACTAACAGAAAATGAGTCT |
| Sequence-based reagent | *cco-1*, forward | This study | qPCR primers | TTTCGGCTATTGTTCGCATT |

*Continued on next page*

*Continued*

| Reagent type (species) or resource | Designation | Source or reference | Identifiers | Additional information |
|---|---|---|---|---|
| Sequence-based reagent | *cco-1*, reverse | This study | qPCR primers | GCCGTCTTAGCA AGTTGAGC |
| Sequence-based reagent | *atp-3*p::ATP-3::GFP | http://www.sunybiotech.com/ | sgRNA target site | Sg1: <u>CCC</u>TTGCCACCG CCATCTAAatt |
| Sequence-based reagent | *atp-3*p::ATP-3::GFP | http://www.sunybiotech.com/ | sgRNA target site | Sg2:<u>CCG</u>CCATCT AAatttttcccaaa |

## Contact for reagent and resource sharing

Further information and requests for resources and reagents should be directed to and will be fulfilled by the Lead Contacts, Gordon Lithgow, glithgow@buckinstitute.org, Julie Andersen, jandersen@buckinstitute.org, and Suzanne Angeli, suzanne.angeli@gmail.com.

## Nematode and bacterial culture conditions

Nematodes were maintained on nematode growth medium (NGM) plates. NGM plates were seeded with *Escherichia coli* OP50 obtained from CGC that was grown in LB broth at 37°C for 18 hr shaking at 225 rpm. Plates with bacteria were dried for 48 hr before use.

For RNAi experiments, *E. coli* HT115 (DE3) bacteria obtained from the Ahringer and Vidal RNAi Library were used (*Kamath et al., 2003*; *Rual et al., 2004*). All RNAi clones were verified via sequencing (Eurofins). RNAi plates were prepared by cooling NGM to 55°C and supplementing with a final concentration of 50 µg/ml carbenicillin and 1 mM Isopropyl β-d-1-thiogalactopyranoside (IPTG). RNAi bacteria were inoculated with one colony of RNAi bacteria into LB with 50 µg/ml carbenicillin and were grown shaking overnight for 18 hr at 37° at 225 rpm.

## Post-developmental timing

To achieve synchronous nematode populations, day 1 adult nematodes were allowed to lay eggs for 2 hr on seeded NGM plates. For convenience, nematodes were developed at 25°C on *E. coli* OP50 until worms were visibly past the L4 stage (loss of crescent) but not yet gravid, approximately 45 hr for wild-type (although the time it takes for the worms to reach the young adult stage varies by strain). Nematodes were shifted to 20°C once they reached adulthood.

## RNAi treatment

For developmental treatments, synchronized eggs were moved onto plates seeded with RNAi bacteria and developed at 20°C for 72 hr. Nematodes were then either collected for analysis or for lifespans, remained on RNAi bacteria for the remainder of their life for survival analysis. For post-developmental treatments, synchronized eggs were developed on plates with *E. coli* OP50 at 25°C until the young adult stage and then transferred to RNAi plates at 20°C. Nematodes were collected after 48 hr for analysis or for lifespans, moved onto *E. coli* OP50 for the remainder of their life.

## Quantitative RT-PCR

Approximately 300 adult nematodes were collected; nematode pellets were resuspended in 300 µL RNA Lysis Buffer and frozen. Pellets was thawed, vortexed, and snap frozen three times. Zymo Research Quick-RNA MiniPrep kit was used to extract RNA.

## Lifespans

Day 1 adult nematodes were allowed to lay eggs for 2 hr on seeded NGM plates to obtain a synchronous aging population. 5-fluoro-2-deoxyuridine (FUdR) was omitted from plates due to its potentially confounding effects (*Angeli et al., 2013*). Worms are transferred to freshly seeded bacterial plates every day for the first 7 days of adulthood and then as needed afterwards. Worms were scored as dead when they failed to respond to gentle prodding with a platinum wire. Worms that experienced matricide or bagging were censored.

## TMRM staining

Plates were prepared by spotting seeded NGM plates with a TMRM solution diluted in water to a final concentration of 0.1 µM in the plates. Water was used as a solvent control. Plates were allowed to dry for 24 hr before use. For developmental experiments, synchronized eggs were placed on plates for 72 hr and then nematodes were collected for analysis. For post-developmental experiments, young adult nematodes were placed on plates for 48 hr and then collected for analysis.

## CsA treatment

Plates were prepared by spotting seeded NGM plates with a CsA (stock solution in DMSO) solution diluted in 100% ethanol. Comparable amounts of DMSO and ethanol were used as solvent controls. Plates were allowed to dry for 24 hr before use. For developmental experiments, synchronized eggs were placed on plates for 72 hr and then nematodes were collected for analysis. For post-developmental experiments, young adult nematodes were placed on plates for 48 hr and then collected for analysis; for lifespans, worms were moved to regular NGM plates for the remainder of their life after 48 hr on drug-treated RNAi bacteria.

## Microscopy

Worms were anesthetized with 2 mM levamisole and mounted on 2% agarose pads on glass slides. Fluorescence micrographs of GFP and TMRM were taken using a Zeiss Imager A2 at 5× magnification with 600 ms exposure using the ZEN software. GFP expression was enhanced using the brightness/contrast tool in Photoshop. The same parameters were used for all images.

Confocal micrographs of mitochondrially targeted GFP and the cytosolic calcium sensor D3cpv (*Zhang et al., 2016*) were taken using a Zeiss LSM780 laser scanning confocal microscope using a 63× Plan Apochromat NA1.4. To visualize outlines of mitochondria, Image Analyst MKII (Image Analyst Software, Novato, CA) was used. Selected rectangular regions of interest (ROIs) from the worm intestine were segmented and converted to outlines by a modification of the 'Segment mitochondria' pipeline. Emission ratio images of D3cpv were excited at 440 nm and captured at 450–490 nm and 520–560 nm and analyzed in Image Analyst MKII. Images were Wiener filtered, and the ratio of the 540 nm over the 470 nm channel, indicative of cytosolic calcium concentration, was calculated and showed in pseudo-color coding. Emission ratios were determined in ROIs in the posterior intestine by the Plot Ratio function.

## Western blot

Approximately 30–50 adult worms were collected in S-basal buffer. Supernatant was removed and nematodes were flash-frozen. Worms pellets were resuspended in 2% SDS sample buffer with 2.5% β−mercaptoethanol and samples were boiled for 10 min. Samples were subjected to SDS-PAGE using 4–12% SDS gels (Invitrogen) and transferred to Immun-Blot PVDF Membrane (Bio-Rad) using Bio-Rad western blot criterion apparatus. Membranes were blocked with 5% non-fat dry milk blocking solution; concentrations for antibodies were 1:1000 for primary antibodies and 1:2000 for secondary antibodies.

## Statistics

Significance between control and experimental groups was determined by using two-tailed Student's *t*-test. Asterisks denote corresponding statistical significance: *$p < 0.05$; **$p < 0.01$; ***$p < 0.0001$. Error bars were generated using the standard error of the mean (SEM), typically from three pooled biological replicates. GraphPad Prism 7 was used to plot survival curves. Log rank (Mantel–Cox) test in Prism was used to determine significance between the control and experimental groups.

## GFP quantification

GFP intensity of worms was quantified using ImageJ 1.52A. The 'integrated density' of GFP expression and length of worms was measured using ImageJ tools. Integrated density value was normalized by number of worms and average length of worms. The final value is in arbitrary units.

## Acknowledgements

We thank members of the Lithgow and Andersen labs for helpful discussion and advice on this research and the Buck Institute Morphology Department for their core services. *C. elegans* strains used in this work were provided by the Caenorhabditis Genetics Center (CGC), funded by the NIH Office of Research Infrastructure Programs (P40OD010440), and the Japanese National BioResource Project. The Y82E9BR.3 RNAi clone was a generous gift from Dr. Seung-Jae V Lee from the Seoul National University College of Medicine. This work was supported by NIH grants RFAG057358 and R01AG029631 and the Larry L. Hillblom Foundation.

## Additional information

### Funding

| Funder | Grant reference number | Author |
|---|---|---|
| National Institute on Aging | R01AG029631 | Gordon Lithgow |
| Larry L. Hillblom Foundation | | Manish Chamoli |
| National Institute on Aging | RF1A057358 | Gordon Lithgow |

The funders had no role in study design, data collection and interpretation, or the decision to submit the work for publication.

### Author contributions

Suzanne Angeli, Conceptualization, Data curation, Formal analysis, Supervision, Validation, Investigation, Visualization, Methodology, Writing - original draft, Project administration, Writing - review and editing; Anna Foulger, Manish Chamoli, Validation, Investigation, Methodology; Tanuja Harshani Peiris, Azar Asadi Shahmirzadi, Investigation, Methodology; Akos Gerencser, Resources, Methodology; Julie Andersen, Gordon Lithgow, Conceptualization, Supervision, Funding acquisition, Writing - original draft, Writing - review and editing

### Author ORCIDs

Suzanne Angeli (iD) https://orcid.org/0000-0002-6229-9648
Manish Chamoli (iD) https://orcid.org/0000-0003-0339-7894
Gordon Lithgow (iD) https://orcid.org/0000-0002-8953-3043

### Decision letter and Author response

Decision letter https://doi.org/10.7554/eLife.63453.sa1
Author response https://doi.org/10.7554/eLife.63453.sa2

## Additional files

### Supplementary files

• Transparent reporting form

### Data availability

Data generated or analyzed during this study are included in the manuscript and supporting files. Worm strains generated from this study will be deposited and available via the Caenorhabditis Genetics Center.

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
