## [Decision Letter]

**Acceptance summary:**

Both reviewers found the manuscript to be of high impact to the field. Particularly intriguing is the finding that disruption of a complex V component activates the mitochondrial unfolded protein response and affects longevity via mechanisms that appear distinct from disruptions of the other complexes, as previously reported.

**Decision letter after peer review:**

Thank you for submitting your article "The mitochondrial permeability transition pore activates a maladaptive mitochondrial unfolded protein response" for consideration by *eLife*. Your article has been reviewed by 2 peer reviewers, and the evaluation has been overseen by a Reviewing Editor and Jessica Tyler as the Senior Editor. The reviewers have opted to remain anonymous.

The reviewers have discussed the reviews with one another and the Reviewing Editor has drafted this decision to help you prepare a revised submission.

Summary:

The authors provided evidence to suggest that RNAi knockdown of the mitochondrial ATP synthase subunit atp-3 in adults triggers mitochondrial permeability transition pore (mPTP), resulting in activation of the mitochondrial unfolded protein response (UPRmt) and shortening of lifespan. This is particularly intriguing as this observation in adult animals contradicts the well-established findings in the field that RNAi of various mitochondrial electron transport chain subunits, including atp-3, during development leads to a transient activation of the UPRmt during larval but not adult stage, and extension of adult lifespan. Further investigation into UPRmt during development vs in adulthood will be important for understanding this important pathway in aging. This study will also contribute to helping clarify the controversial field of mPTP.

Essential revisions:

The reviewers appreciated the potential importance of this work, but more evidence are needed to support the major claims put forth.

A) Additional experimental evidence are required to demonstrate that atp-3 RNAi indeed induces mPTP.

1. The authors stated at the beginning of the manuscript that the opening of the mPTP is characterized by a loss of mitochondrial membrane potential and an increase in cytosolic ca^2+^, however, the measurements of the membrane potential and the cytosolic ca^2+^ levels were missing from many key experiments. For example, in figure S2, the authors showed that the activation of the pdvUPRmt is reversible, however, whether the loss of membrane potential and the increase in cytosolic ca^2+^ were also reversed was not shown. The same measurement should also apply to the situation when the pdvUPRmt induced by atp-3 RNAi was suppressed by the RNAi against the ATPases subunits (a and b), the two UPRmt regulators (haf-1(the mitochondrial protein), atfs^-1^(the downstream TF), or by the pharmacological inhibitors mPTP, and in the long-lived mutants (daf-2, glp-1)).

2. Along with the previous point, the authors showed that loss of FCCP, a mitochondrial uncoupler, did not induce a UPRmt in adults, therefore excluding the possibility that loss of mitochondrial membrane potential is not sufficient to induce pdvUPRmt. However, it has been shown that FCCP treatment induces the UPRmt in mammalian cells (Fessler et al., 2020). Therefore, controls with measurement of the mitochondrial membrane potential and the ca^2+^ measurement after FCCP treatments should be included to demonstrate the efficiency of the drug treatment.

3. It is also difficult to understand the phenotypical differences between RNAi of atp-1 and atp-3. Loss of atp-1 and atp-3 can both induce the pdvUPRmt, however, atp-1 RNAi had no effect on the membrane potential but could partially suppress the pdvUPRmt induced by atp-3 RNAi.

The reviewers pointed out papers in the literature that argued against the proposal that the mPTP is associated with the ATP synthase. Given that this is a controversial topic, clear evidence must be provided to convince the readers otherwise.

B) Epistasis analysis between dvUPRmt and pdvUPRmt will be key to understand the interesting opposing observations.

1) Mechanistically, why the inductions of dvUPRmt and pdvUPRmt have such huge differences in the regulation of organismal aging? The authors have not excluded the possibility that the mPTP is not formed when dvUPRmt is induced.

2) Inhibition of multiple OXPHOS components such as CCO-1 during development activates UPRmt and extends worm lifespan. However, inhibition of CCO-1 at or after the L4 stage does not activate UPRmt or extend lifespan. Here, the authors demonstrate that inhibition of atp-3/OSCP during development activates the UPRmt and extends lifespan. However, inhibition of atp-3/OSCP at the young adult stage also activates the UPRmt, but shortens lifespan. Which is dominant? Is cco-1(RNAi)-dependent lifespan extension suppressed by atp-3(RNAi) administered at the young adult stage?

This line of investigation is necessary to provide a clearer picture about how activation of UPRmt influences aging.

C) The data on genetic interactions with other longevity mutants appear premature. The reviewers suggested completely removing that part from this manuscript, so that the current manuscript can focus on A and B above.

– The section titled "Longevity mutants display exceptional mitochondria partially via FOXO/daf-16" is intriguing but much too underdeveloped. I agree that it is interesting that several longevity mutants suppress the post-developmental UPRmt.

– However, the subsequent claims about "exceptional mitochondria" and increased mitochondrial mass using mtGFP as a marker are not convincing. For example, the ges^-1^ promoter is induced by DAF-16, confounding the results. Mitochondriall staining independent of mtGFP should be performed such as TMRE. The imaging should also be of higher magnification. mtDNA quantification should be provided as an imaging-independent indicator of mitochondrial mass. The authors should also determine if the "exceptional mitochondria" in these worms provide more mitochondrial activity by measuring oxygen consumption etc. The data suggesting that mitophagy contributes to the loss of mitochondrial mass are also underdeveloped. Images should be provided and mitochondrial mass should be quantified via a mtGFP independent mechanism. I would suggest simply removing this section as it largely detracts from the rest of the manuscript.

---

## [Author Response]

Essential revisions:The reviewers appreciated the potential importance of this work, but more evidence are needed to support the major claims put forth.A) Additional experimental evidence are required to demonstrate that atp-3 RNAi indeed induces mPTP.

To address this critique, we decided to use the well characterized mPTP inhibitior, cylosporin A (CsA), to determine if it could reverse the mPTP characteristics caused by adult loss of OSCP*/atp-3.* The use of CsA is an established method to help confirm the identity of the mPTP in many models, such as isolated mitoblasts and cell lines. We found that CsA was able to reverse all mPTP-like characteristics due to adult loss of OSCP/*atp-3*, which included loss of MMP, rise in cytosolioc ca^2+^, lifespan shortening, and mitochondrial rounding/fragmentation (Figures 1C, 1F, 1I, 1J, 3G). These finding significantly strengthen our claim that our observations are due to the mPTP.

To additionally address this critique, we further examined our epistasis experiments that showed that the pdvUPR^mt^ can be suppressed due to loss of additional F-ATP synthase subunits. Epistatic experiments also reverse nearly all mPTP-like characteristics due to adult loss of OSCP/*atp-3*, which include loss of MMP, rise in cytosolioc ca^2+^, lifespan shortening, and mitochondrial rounding/fragmentation (Figure 6). These finding support a model in which F-ATP synthase, especially the proton-driving c-ring rotor, is likely to form the mPTP.

1. The authors stated at the beginning of the manuscript that the opening of the mPTP is characterized by a loss of mitochondrial membrane potential and an increase in cytosolic ca^2+^, however, the measurements of the membrane potential and the cytosolic ca^2+^ levels were missing from many key experiments. For example, in figure S2, the authors showed that the activation of the pdvUPRmt is reversible, however，whether the loss of membrane potential and the increase in cytosolic ca^2+^ were also reversed was not shown. The same measurement should also apply to the situation when the pdvUPRmt induced by atp-3 RNAi was suppressed by the RNAi against the ATPases subunits (a and b), the two UPRmt regulators (haf-1(the mitochondrial protein), atfs^-1^(the downstream TF), or by the pharmacological inhibitors mPTP, and in the long-lived mutants (daf-2, glp-1)).2. Along with the previous point, the authors showed that loss of FCCP, a mitochondrial uncoupler, did not induce a UPRmt in adults, therefore excluding the possibility that loss of mitochondrial membrane potential is not sufficient to induce pdvUPRmt. However, it has been shown that FCCP treatment induces the UPRmt in mammalian cells (Fessler et al., 2020). Therefore, controls with measurement of the mitochondrial membrane potential and the ca^2+^ measurement after FCCP treatments should be included to demonstrate the efficiency of the drug treatment.

We verified that FCCP did induce a loss of MMP during adulthood, demonstrating the drug efficacy, while also failing to increase cytosolic ca^2+^ levels (Figure 1—figure supplement 1A, B), demonstrating that loss of MMP per se is not sufficient to recapitulate the mPTP.

3. It is also difficult to understand the phenotypical differences between RNAi of atp-1 and atp-3. Loss of atp-1 and atp-3 can both induce the pdvUPRmt, however, atp-1 RNAi had no effect on the membrane potential but could partially suppress the pdvUPRmt induced by atp-3 RNAi.

We further tested the role of the a/*atp-1* subunit. In addition to its loss not inducing a loss of MMP, its loss also did not lead to a rise in cytosolic ca^2+^ (Figure 1E) and had minor effects on lifespan (Figure 4F). Thus, its ability to induce a moderately robust pdvUPR^mt^ does not necessitate that it display mPTP characteristics. We conclude from these findings that mPTP activation is very specific to the loss of OSCP/*atp-3*.

The reviewers pointed out papers in the literature that argued against the proposal that the mPTP is associated with the ATP synthase. Given that this is a controversial topic, clear evidence must be provided to convince the readers otherwise.

We added more context in the introduction that references that experts in the field of the mPTP largely consider F-ATP synthase to be the probable pore-forming component. This is not without dispute though, as we also further discuss in the introduction. Thus, while our findings do not conclusively ascertain that the F-ATP synthase is the mPTP, they are supportive of the prevailing F-ATP synthase models of the mPTP.

B) Epistasis analysis between dvUPRmt and pdvUPRmt will be key to understand the interesting opposing observations.1) Mechanistically, why the inductions of dvUPRmt and pdvUPRmt have such huge differences in the regulation of organismal aging? The authors have not excluded the possibility that the mPTP is not formed when dvUPRmt is induced.

We performed additional experiments analysing cytosolic ca^2+^ release and found that loss of OSCP/*atp-3* during development does not lead to a release of cytosolic ca^2+^ (Figure 1—figure supplement 1D), a canonical feature of the mPTP. This finding, in addition to our previous findings that the dvUPR^mt^ is not responsive to CsA or genetic mPTP factors (Figure 3A, B), lead us to conclude that the dvUPR^mt^ is not likely coupled to the mPTP.

2) Inhibition of multiple OXPHOS components such as CCO-1 during development activates UPRmt and extends worm lifespan. However, inhibition of CCO-1 at or after the L4 stage does not activate UPRmt or extend lifespan. Here, the authors demonstrate that inhibition of atp-3/OSCP during development activates the UPRmt and extends lifespan. However, inhibition of atp-3/OSCP at the young adult stage also activates the UPRmt, but shortens lifespan. Which is dominant? Is cco-1(RNAi)-dependent lifespan extension suppressed by atp-3(RNAi) administered at the young adult stage?This line of investigation is necessary to provide a clearer picture about how activation of UPRmt influences aging.

This interesting experiment proposed by the reviewers led to an interesting result. We found that developmental treatment with COX5B/*cco-1* RNAi followed by post-developmental treatment with OSCP/*atp-3* RNAi did not significantly alter the long-lived phenotype cause by developmental *cco-1*, suggesting that the *cco-1* regulation of longevity is “dominant”. We agree further investigation into how the dvUPR^mt^ and the pdvUPR^mt^ are regulated is an area of high priority for future studies.

C) The data on genetic interactions with other longevity mutants appear premature. The reviewers suggested completely removing that part from this manuscript, so that the current manuscript can focus on A and B above.

We agree and have accordingly removed this portion.